# Cooperation and partner choice among Agta hunter-gatherer children: An evolutionary developmental perspective

**Daniel Major-Smith**[1,2]*, **Nikhil Chaudhary**[3], **Mark Dyble**[2], **Katie Major-Smith**[4], **Abigail E. Page**[5], **Gul Deniz Salali**[2], **Ruth Mace**[2], **Andrea B. Migliano**[6]

1 Population Health Sciences, Bristol Medical School, University of Bristol, Bristol, United Kingdom, 2 Department of Anthropology, University College London, London, United Kingdom, 3 Department of Archaeology, Leverhulme Centre for Human Evolutionary Studies, University of Cambridge, Cambridge, United Kingdom, 4 Department of Business and Social Sciences, Plymouth Marjon University, Plymouth, United Kingdom, 5 Department of Population Health, London School of Hygiene and Tropical Medicine, London, United Kingdom, 6 Department of Anthropology, University of Zurich, Zurich, Switzerland

* dan.smith@bristol.ac.uk

**Data Availability Statement:** Data and analysis code are openly-available on DM-S's GitHub page: https://github.com/djsmith-90/AgtaChildCoop.

## Abstract

Examining development is essential for a full understanding of behaviour, including how individuals acquire traits and how adaptive evolutionary forces shape these processes. The present study explores the development of cooperative behaviour among the Agta, a Filipino hunter-gatherer population. A simple resource allocation game assessing both levels of cooperation (how much children shared) and patterns of partner choice (who they shared with) was played with 179 children between the ages of 3 and 18. Children were given five resources (candies) and for each was asked whether to keep it for themselves or share with someone else, and if so, who this was. Between-camp variation in children's cooperative behaviour was substantial, and the only strong predictor of children's cooperation was the average level of cooperation among adults in camp; that is, children were more cooperative in camps where adults were more cooperative. Neither age, sex, relatedness or parental levels of cooperation were strongly associated with the amount children shared. Children preferentially shared with close kin (especially siblings), although older children increasingly shared with less-related individuals. Findings are discussed in terms of their implications for understanding cross-cultural patterns of children's cooperation, and broader links with human cooperative childcare and life history evolution.

## Introduction

Cooperation is a key human adaptation. Despite much research investigating how cooperative behaviour develops over childhood in Western societies [1–3], relatively little developmental work has occurred in the context of cooperation in small-scale societies, and hunter-gatherers in particular. Developmental research is necessary for a complete understanding of behaviour [4], while cross-cultural research, especially in small-scale societies, is required to assess how

**Funding:** This work was supported by the Leverhulme Trust (grant no. RP2011 R045; awarded to ABM and RM). DM-S was also supported by the John Templeton Foundation (grant ID: 61917). The funders had no role in study design, data collection and analysis, decision to publish, or preparation of the manuscript

**Competing interests:** The authors have declared that no competing interests exist.

behaviour develops outside of Western societies, where behaviour may not be representative of current human diversity or our deeper evolutionary history [5]. This ontogenetic perspective may be especially valuable for understanding human cooperation, as even though children are somewhat dependent on provisioning until adulthood [6], they are not merely burdens to adults as they engage in useful cooperative activities such as caring for other children, performing domestic tasks and foraging [7–10].

Although spontaneous helping occurs in children by the age of 12 months (reviewed in [2]), previous research on the development of cooperative behaviour has found that levels of cooperation tend to increase throughout childhood. This has been found both in Western [11] and small-scale [12–15] societies, although for an exception see [16]. For instance, across six diverse societies, including hunter-gatherer, horticultural and urban populations, patterns of costly cooperation were similar in all populations until middle childhood, with levels of cooperation decreasing between the ages of 3 and 6–8 years, after which societies diverged to approximate population-specific adult levels of cooperation [12] (see also [15]). One explanation for this pattern of results is that middle childhood may be a key time when children become more sensitive to the local social norms and alter their cooperative behaviour accordingly [12,15,17]. Thus, in addition to cooperation increasing over childhood, previous research suggests that children may increasingly attune their cooperative behaviour as they develop to correspond to adult patterns of cooperation [12].

However, the majority of these studies have used experimental methods which either paired children with anonymous others (e.g., [11,15]) or with unrelated individuals (although partners were known to each other, such as attending the same school or living in the same village; e.g., [12,13,16]). These studies may overlook the important role of 'partner choice'–who children choose to interact and cooperate with–in determining cooperative behaviour [18]. This may be important for two reasons. First, who children are interacting with is likely to impact how much they cooperate, meaning that ignoring the role of partner choice may confound assessments of overall levels of cooperation. For instance, young children may preferentially only share with people they know, while older children may be more likely to share with strangers. In experimental games with anonymous recipients, older children may appear more cooperative than younger children, but this may be due to partners being anonymous, rather than differences in overall levels of cooperation. By only assessing cooperation towards non-kin or strangers, many previous studies may have overlooked how partner choice shapes children's cooperative behaviour.

A second reason for considering partner choice is that it allows an exploration of who children share with and whether this changes over time. For instance, following kin selection theory [19], children–all else being equal–may be more likely to cooperate with close kin, rather than non-kin or more distant relatives, because of the indirect fitness benefits accrued by helping close kin grow, survive and reproduce. Experimental studies have demonstrated that North American [20] and Chinese [21] children are more likely to share with kin and friends, relative to strangers, although for an exception to this kin-bias among French children, see [22]. In an observational study of food-sharing with Hadza children (Tanzanian hunter-gatherers), resources were preferentially directed towards reciprocal partners and kin [14]. Other studies in small-scale societies have shown that children preferentially invest in close kin, especially siblings, over more distant kin [7,23,24]. Numerous studies of adult cooperation in small-scale societies have also shown that relatedness is a strong predictor of cooperation [25–27]. Thus, partner choice likely shapes patterns of cooperation among children, particularly regarding kinship which we focus on in this study.

Patterns of partner choice are unlikely to remain static during development. Among two forager populations, the Filipino Agta (the current study population) and the Congolese

BaYaka, interactions with non-kin increased throughout childhood [28]. This potentially reflects the need to establish extensive non-kin networks of cooperation in adulthood [29–31]. Children also need to learn the social and economic skills necessary for independent living, including foraging, mate acquisition and alliance formation [6,8,32]. Learning these skills is likely to require extensive interactions with peers of a similar age, especially in playgroups [9,33,34], rather than learning solely from siblings [17,35]. Collectively, these considerations lead to the broad expectation that cooperation will be directed towards close kin in early childhood, while social and cooperative networks will broaden throughout childhood, including increasing interactions with non-kin. This generalised prediction is predominantly based on our current knowledge of childhood, kinship and cooperation in mobile immediate-return hunter-gatherers, and it is likely that these patterns will vary depending on the local social, ecological and cultural conditions, particularly in non-foraging societies. For instance, children's involvement in subsistence activities varies across populations [36], as does the amount of care provided by siblings [7,37]. In many Western and industrialised societies, meanwhile, children have few siblings and institutions such as nurseries and schools bring large numbers of unrelated children together from a young age [38]. Nonetheless, as a broad prediction we expect that the importance of relationships with non-kin will increase through childhood. Therefore, to compare overall levels of cooperation which are not confounded by constraining children to certain partners, as well as to explore who children share with and whether this changes with age, incorporating partner choice into experimental designs is necessary. While experimental designs which incorporate partner choice are increasingly employed among adults [26,39–42], there are few studies applying such methods with children, especially in small-scale societies.

Using a cross-sectional experimental design, we aim to investigate children's cooperative behaviour among the Agta, a population of Filipino hunter-gatherers. Our design allows children to share with multiple partners, permitting an exploration of both how much children share and who they share with. This research asks the following questions: i) Without constraining who children interact with, how do levels of cooperation develop over childhood? and ii) How do patterns of partner choice change throughout childhood, with a focus on genetic relatedness? Based on existing research we predict that levels of cooperation will increase with age. We also expect that this developmental trajectory will vary between camps, with cooperation in later childhood matching camp-specific adult-levels of cooperation, as previous research has identified substantial variation in cooperation between Agta camps using similar games conducted among adults [26]. Regarding the second question, consistent with kin selection theory we predict that much cooperation will be directed towards close kin, especially siblings. Given the necessity of developing alliances and skills for adulthood, we also expect that this kin-bias will decrease with age, as children increasingly interact and cooperate with distant kin and non-relatives.

## Methods

### Ethnographic background

The Agta are a population of indigenous hunter-gatherers who reside in, and are nominally responsible for co-managing [43], the Northern Sierra Madre Natural Park in Isabela Province, the Philippines [44]. Two sub-populations were the focus of this study: the Palanan Agta, who number ~1,000 individuals, and the Maconacon Agta, who number ~250 individuals. These sub-populations are largely separate as they live approximately 50 kilometres apart from one another and share few genealogical links. These Agta populations live predominantly by foraging, especially fishing, but also collecting honey, gathering wild plants, and, to a lesser degree, hunting. Many foraged resources are traded with the local agricultural non-Agta

population for rice and other food or commodities. The Agta are increasingly engaged in agricultural work and wage labour [45], often consisting of tending rice fields, harvesting crops, clearing land and housekeeping. Over the total Palanan Agta population, approximately 75% of time engaged in subsistence activities is devoted to foraging, with the remaining 25% spent on agriculture and wage labour. The average camp size is 7 houses (49 individuals), but this varies substantially, from camps consisting of a single dwelling (7 individuals) to large camps of 26 houses (156 individuals). Mobility is high as individuals frequently move camp, although this is variable as some individuals and camps are less mobile than others [26].

Agta children are breastfed on-demand until around two years of age, after which the amount of time and care received from their mother reduces and they increase their participation in mixed-age and -sex playgroups. In these playgroups they engage in games, role-play and 'work-play' (where foraging is integrated into play; [9]). From the age of five or six, children—and particularly girls—become increasingly involved in the household economy, providing childcare to younger siblings (often within the playgroup) and completing domestic tasks [9,46]. As children get older, their time spent foraging increases, especially for boys [45]. School attendance is also highly variable and largely depends on the distance to the nearest population centre.

Initial fieldwork to collect demographic data was conducted between April and June 2013, while additional fieldwork and conducting these experimental games took place between February and October 2014. Ethical clearance for the project was granted by the University College London ethics review board (UCL Ethics Code 3086/003) and carried out with permission from local government and Agta leaders. Informed consent for all children involved in this study was obtained from all parents/guardians after group and individual consultation, with an explanation of the research in the local language.

## Data collection

Games were conducted with 179 Agta children between the ages of 3 and 18 (mean = 8.9; SD = 3.2; S1 Fig in S1 File), of whom 87 (48.6%) were male. Ages for most children were unknown, so were robustly estimated based on the Bayesian procedure described in [47], and briefly summarised in supplemental section S1. Data were collected from a total of 14 camps (S1 Table in S1 File).

To assess levels and patterns of cooperation, a simple resource sharing game was developed. In this game, children were told that there were five small individually-wrapped candies, and for each one they had to decide: i) if they wanted to keep the candy for themselves or whether to give it to somebody else; and ii) if they wanted to give the candy away, to whom. Children were shown each candy sequentially (i.e., one-at-a-time). This design was chosen to be as simple as possible for children to understand, without the need for complex apparatus or detailed explanation. There were no restrictions on age or whether recipients had to be from the same camp, although few children gave to adults (see below) and no-one gave to anyone from a different camp. Children could also nominate the same recipient repeatedly if they wished. This design permitted an exploration of both levels of cooperation (total amount shared) and patterns of partner choice (who they shared with). As discussed in the introduction, both of these are decisions made repeatedly by children in their daily lives—e.g., sharing food, looking after younger children—and are essential for a full understanding of cooperative behaviour.

Games were conducted in private, out of sight and earshot of other children, with just the child, experimenter and translator. Children were told that their decisions were secret, so nobody else would know how much they kept or who they gave to, and that there were no right or wrong answers. Before playing the game, children were asked if they understood the

rules and if they had any questions. After this, the game was played. The total number of candies earned was given to children after all children participated in the game, which included both the amount each child kept for themselves and resources given to them by other children. To ensure that every child received something, all children were also given additional candies; children playing the games were not aware of these additional candies at the time. Games were generally played towards the end of our visit to each camp, meaning that gifts were normally given out within a few days of the child participating. The game was played with as many children as possible in each camp using opportunistic sampling, although some children did not take part because they were too shy and/or too young to fully understand the game, or just did not want to. For additional details of the study methods, see supplemental section S2.

## Statistical analysis

Analyses were conducted using *R* version 4.0.4 [48]. We used the package 'brms' to fit Bayesian multi-level models using Stan [49]. Unless otherwise stated, all analyses used non-informative default priors and consisted of 4 chains, each with 4,000 iterations, of which 1,000 were warm-up iterations, giving a total of 12,000 posterior samples. Visual inspection and r-hat statistics were used to ensure convergence between chains and that all parameter estimates had stabilised (with an r-hat of 1.00 indicating convergence). Model fit was assessed using leave-one-out cross-validation (LOO). Analyses relating to overall levels of cooperation will be described first, with the partner choice analyses described afterwards.

**Amount shared analyses.** The response variable was the total number of resources given to others, between 0 and 5. Predictor variables included: age (in years), sex (child's assigned sex at birth, based on parental genealogical interviews; 0 = female; 1 = male), average relatedness of the focal child to all other children in camp (Wright's coefficient of relatedness; derived based on genealogical interviews and calculated using the 'kinship2' package [50]), and the average amount of resources given to others by adults in camp (using data from similar games conducted with adults [26]; see supplemental section S3 for a description of this adult game).

As we were interested in estimating how each predictor variable was associated with the outcome (amount shared), to understand whether these effects can be given a potentially causal interpretation the causal relations between variables were considered. A Directed Acyclic Graph (DAG) representing these assumed casual relationships is displayed in S2 Fig in S1 File (for primers on DAGs and causal inference, see [51,52]). As nothing can cause age or sex, unadjusted estimates of these effects are likely to be unbiased. We also assume that relatedness between children is also independent of both other predictor variables and potential confounders. Finally, based on previous research with Agta adults we assume that adult levels of cooperation are shaped by factors such as camp stability/repeated interactions, resource availability/need, and storytelling [26,53]. It is possible that these factors directly shape children's levels of cooperation, rather than being mediated through adult levels of cooperation; however, here we are interested in whether adult levels of cooperation are associated with child levels of cooperation, regardless of whether this is through adult cooperation directly or through factors which shape both adult and child levels of cooperation. Our hypothesised causal structure suggests that each of our predictor variables may be largely independent from one another.

To control for the hierarchical nature of the data, a multi-level modelling approach was adopted, with children nested within households within camps. Comparison of intercept-only linear multi-level models with different random effects structures indicated that the inclusion of camp ID, but not household ID, improved model fit (see associated data and code). All models regarding the amount shared therefore included camp as a random effect. Analyses were first conducted for each predictor variable in a univariable model, followed by a

multivariable model which included all predictor variables. Given the hypothesised DAG (S2 Fig in S1 File), which assumes that these variables are independent, the univariable and multivariable models ought to provide similar estimates.

When including average adult levels of cooperation as a covariate, only the mean value was used. However, this approach ignores variation in this value (*cf.* [15]). We therefore conducted sensitivity analyses to explicitly incorporate this variation and test the robustness of our results. For univariable and multivariable analyses involving this adult cooperation covariate, we generated 1,000 datasets sampling the adult level of cooperation in each camp from a normal distribution given the camp mean and standard error. Analyses were then conducted on these 1,000 datasets and combined together using the command 'brm_multiple'. To save processing time, we analysed each dataset using 1 chain and a follow-up time of 1,000 iterations. As the number of resources given could only vary between 0 and 5, additional robustness checks were performed using Poisson models (which are suitable for count data) and ordinal regression models (using 'amount given' as an ordered categorical variable). Given the small number of children sharing 5 resources, for the ordinal models children sharing 4 and 5 candies were grouped together. None of these methods are ideal; linear methods because there are only six possible integer responses, Poisson models because of excess '0's in the data, and ordinal regressions because the outcomes are integers rather than ordered categories. However, if all methods provide a similar answer this increases confidence that the results are robust.

Three further sets of analyses were conducted. The first additional set of analyses assessed whether the association between age and cooperation was non-linear, and whether there were different age trajectories by camp. To test this, we created four models: i) linear age fixed effect with camp-level random intercepts; ii) quadratic age fixed effect with camp-level random intercepts; iii) linear age fixed effect with camp-level random slopes and intercepts; and iv) quadratic age fixed effect with camp-level random slopes and intercepts. Models were compared using LOOIC (leave-one-out information criteria).

The second set of further analyses were univariable analyses exploring whether mother, father or average-parental levels of cooperation predicted children's levels of cooperation. Three separate models were conducted; a mother-only model ($n = 155$), a father-only model ($n = 145$), and a combined parental model using the average of the parents' scores (if only one parent participated, their data was used as an average score in the combined parental model; $n = 162$).

Finally, to examine the size of children's personal sharing networks, additional models were performed assessing factors associated with the number of unique recipients that the participant shared with, using the same predictor variables as above. This may differ from the total amount shared; for instance, if a child shared four gifts but gave them all to one sibling, the total number of gifts shared would be '4', while the number of unique recipients would be '1'.

**Sharing partners analyses.** These sharing models are 'gift-based' so only included individuals who shared resources. That is, each gift donated is a separate data point. In our models we used participant ID as a random effect to control for repeated nominations by the same child. Camp-level random effects either resulted in no or minimal improvement in model fit. While recipient ID was considered as an additional potential random effect, there were often major convergence and estimation issues with high r-hat values (>3) indicating poor model fit when including recipient ID, likely because participant and recipient IDs are highly collinear (i.e., most recipients were only nominated by one participant). To simplify the random effects structure, we therefore present results here with just participant ID as a random effect, which captures the majority of non-independence in the data. Where feasible, inclusion of additional camp and/or recipient ID random effects provided similar results, indicating that the results reported here are likely to be robust (see associated data and code). This approach was chosen

over a network analysis, such as the Social Relations Model [52,54], predominantly because many of the recipients did not participate in the games, either because they were too young–many recipients were infants–or chose not to take part. Given this data structure, such network analyses would not be possible.

Of the 320 gifts given by children to others, 12 were to adults over the age of 20 (3.8% of all gifts). Children who shared with adults tended to be older (mean = 12.3; range = 6–18), and most of these gifts– 10 of 12 –were to parents or adult siblings. In order to compare interactions between just children, in addition to removing these adult outliers, all recipients over the age of 20 were omitted, resulting in 308 gifts from 125 children.

For the sharing partners analyses two sets of models were constructed, depending on the response variable of interest. The first analysis assessed whether the relatedness between participant and recipient was greater than expected by chance, based upon background levels of relatedness between children in camp, with 'relatedness' as the outcome and a binary predictor variable indicating whether relatedness referred to recipients or the camp average. In this analysis, a positive coefficient indicates higher relatedness between participant and nominee compared to participant and the camp average.

The second set of analyses explored whether the age or sex of the participant influenced cooperation towards relatives. Two models were constructed with relatedness between participant and recipient as the response variable: one with participant age and sex as predictors, and another with an interaction term between these covariates to explore whether these associations differed by age and sex. To further examine patterns of children's cooperation, supplementary analyses were performed to explore how participant's age, sex and relatedness to recipient were associated with: i) recipient's age; ii) the age difference between participant and recipient (calculated by subtracting participant age from recipient age, with negative values meaning recipient was younger than participant); iii) recipient's sex; and iv) whether participant and recipient were of the same sex. For each of these analyses we began with a model including only main effects of participant's age, sex and relatedness to recipient, and then assessed whether any interaction effects were present. Linear models were used when relatedness and age were the response variables, while logistic models were used when sex was the response variable.

**Table 1. Total number of candies given to others and the number of unique recipients the child shared with (*n* = 179).**

| Amount given to others | Count (%) |
|---|---|
| 0 | 53 (29.6%) |
| 1 | 19 (10.6%) |
| 2 | 41 (22.9%) |
| 3 | 48 (26.8%) |
| 4 | 13 (7.3%) |
| 5 | 5 (2.8%) |
| **Number of unique recipients** | **Count (%)** |
| 0 | 53 (29.6%) |
| 1 | 33 (18.4%) |
| 2 | 43 (24.0%) |
| 3 | 37 (20.7%) |
| 4 | 10 (5.6%) |
| 5 | 3 (1.7%) |

For example, 41 children shared two candies, while 43 children shared with two unique recipients.

## Results

### Amount shared

The mean number of resources shared by children was 1.8 (SD = 1.44), with modes at 0 and 3 (Table 1). Substantial camp-level variation was observed (Fig 1), from a maximum camp average of 3.3 (SD = 0.58, *n* = 3) to a minimum camp average of 0 (SD = 0, *n* = 9), meaning that no-one shared with anyone (interestingly, this was also the camp where no adults shared any resources either). In the intercept-only multi-level model, 29% (95% credible interval [CI] = [11%; 54%]) of the variance in children's cooperation occurred at the camp level.

Next, linear multi-level models were performed on each of the four predictor variables in univariable models, and then together in a multivariable model (Table 2). In this multivariable model the assumption of normality is met (S3 Fig in S1 File) but equal variances (homoskedasticity) may be slightly violated (S4 Fig in S1 File). Consistent with the assumptions encoded in our DAG (S2 Fig in S1 File), there was little difference in the coefficients between the univariable and multivariable models, suggesting that these variables are largely independent.

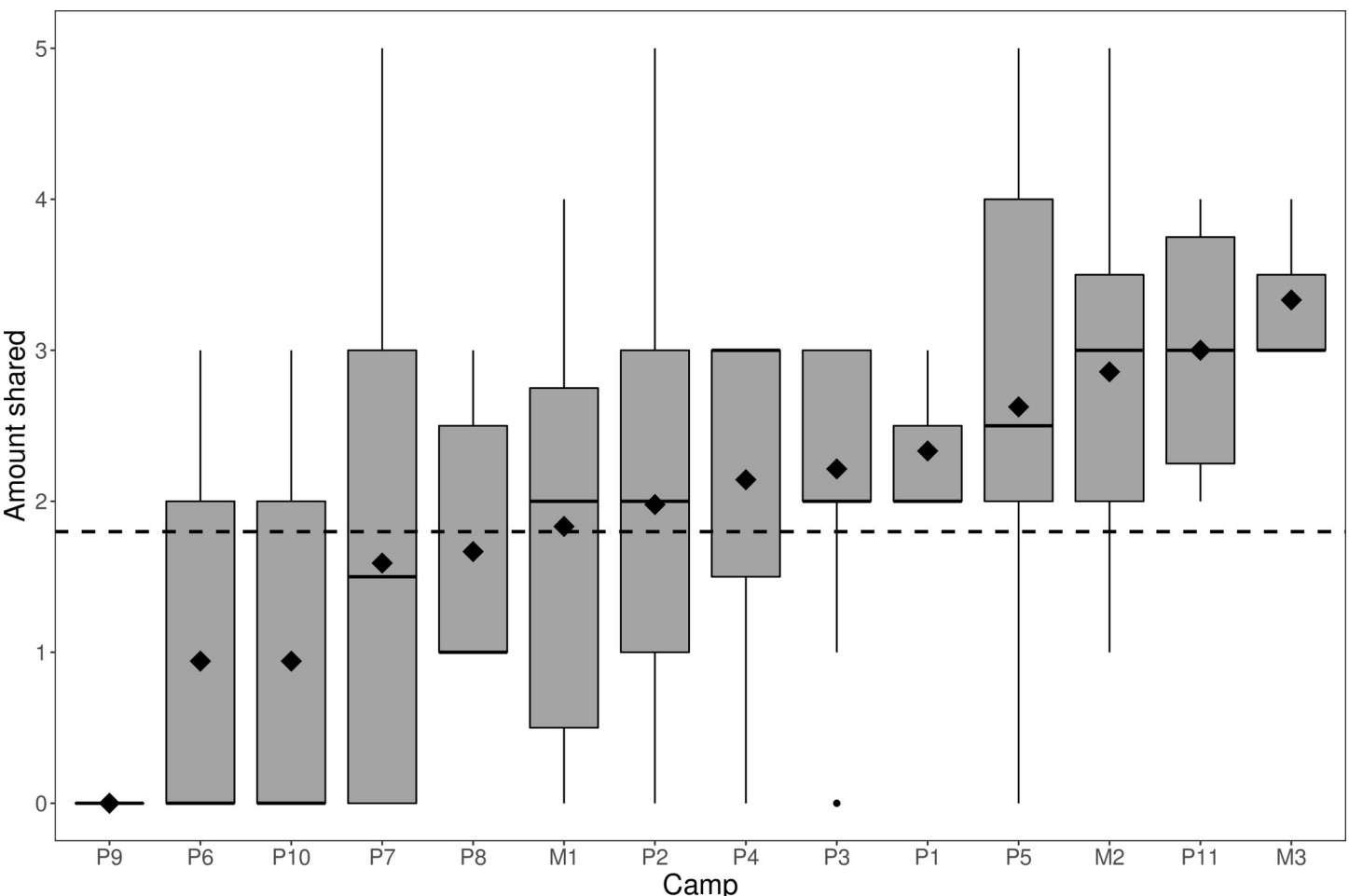

**Fig 1. Box-plot displaying the number of resources given, grouped by camp (*n* = 179; camps = 14).** Boxes represent inter-quartile ranges with the black lines within bars indicating the median. Lines extending above and below boxes display upper and lower quartile ranges. Diamonds indicate the mean value for said camp, while the horizontal dashed line indicates the mean for the whole population.

The only factor strongly associated with children's cooperation was the average adult level of cooperation in camp ($b\_adult_{\text{full model}}$ = 0.031, 95% CI = [0.011; 0.051]), with children from camps where adults were more cooperative displaying greater cooperation (Fig 2). For context, given a 70-percentage point increase in camp-level adult cooperation, which is the difference between the most and least cooperative camps, children would be predicted to share approximately two additional resources. There was, at best, a weak positive association between age and children's cooperation ($b\_age_{\text{full model}}$ = 0.051, 95% CI = [-0.014; 0.116]; S5 Fig in S1 File and Table 2). Results were qualitatively identical when incorporating variation in adult levels of cooperation ($b\_adult_{\text{uni model}}$ = 0.033, 95% CI = [0.014; 0.051]; $b\_adult_{\text{full model}}$ = 0.031, 95% CI = [0.011; 0.051]) and using Poisson or ordinal regression methods (S2 Table in S1 File).

The inclusion of non-linear age fixed effects or random slopes by camp did not improve model fit, suggesting that the age effect was broadly linear and did not vary substantially by camp (Table 3).

Mother ($b$ = 0.007, 95% CI = [-0.004; 0.017]), father ($b$ = -0.005, 95% CI = [-0.015; 0.005]), and joint-parental ($b$ = -0.001, 95% CI = [-0.012; 0.010]) cooperation terms were not associated with child levels of cooperation. Comparable results were found using Poisson and ordinal regression models (S3 Table in S1 File).

While a negligible effect of age on overall levels of children's cooperation was reported, additional models using 'number of unique recipients' did find an effect of age, with older children sharing with a greater number of individuals ($b\_age_{\text{full model}}$ = 0.093, 95% CI = [0.036; 0.151]; S5 Fig and S4 Table in S1 File). Tests of model assumptions are provided in S6 and S7 Figs in S1 File. Given a 10-year increase in age, children were expected to share with approximately one additional person. Together, these models suggest that although children may not have shared more with age, older children did share with a greater number of recipients. As with the previous analysis regarding overall levels of cooperation, we again found that average adult levels of cooperation were strongly associated with an increased number of recipients, while no associations were reported for sex or relatedness (S4 Table in S1 File). Incorporating variation in the mean adult levels of cooperation into consideration returns comparable results and these results are robust to using Poisson or ordinal regression methods (S4 Table in S1 File). There was no evidence for a non-linear age trajectory with number of unique recipients, or that the association with age varied by camp (Table 3).

**Table 2. Results of linear multi-level regression models regarding the factors associated with the number of candies given by children ($n$ = 179, camps = 14).**

| Variable | Level | Univariable models | | Multivariable model | |
|---|---|---|---|---|---|
| | | *Parameter estimate (SE)* | *95% credible intervals* | *Parameter estimate (SE)* | *95% credible intervals* |
| Age | Individual | 0.057 (0.032) | -0.006; 0.120 | 0.051 (0.033) | -0.014; 0.116 |
| Sex (1 = male) | Individual | 0.289 (0.199) | -0.109; 0.680 | 0.271 (0.201) | -0.122; 0.659 |
| Relatedness | Individual | -1.028 (2.528) | -5.980; 3.883 | -0.846 (2.362) | -5.544; 3.685 |
| Adult coop | Camp | 0.032 (0.009) | 0.014; 0.050 | 0.031 (0.010) | 0.011; 0.051 |

Both the univariable (separate) and multivariable (combined) model results are presented here. Positive parameter estimates indicate an increase in the number of resources given to others. SE = standard error.

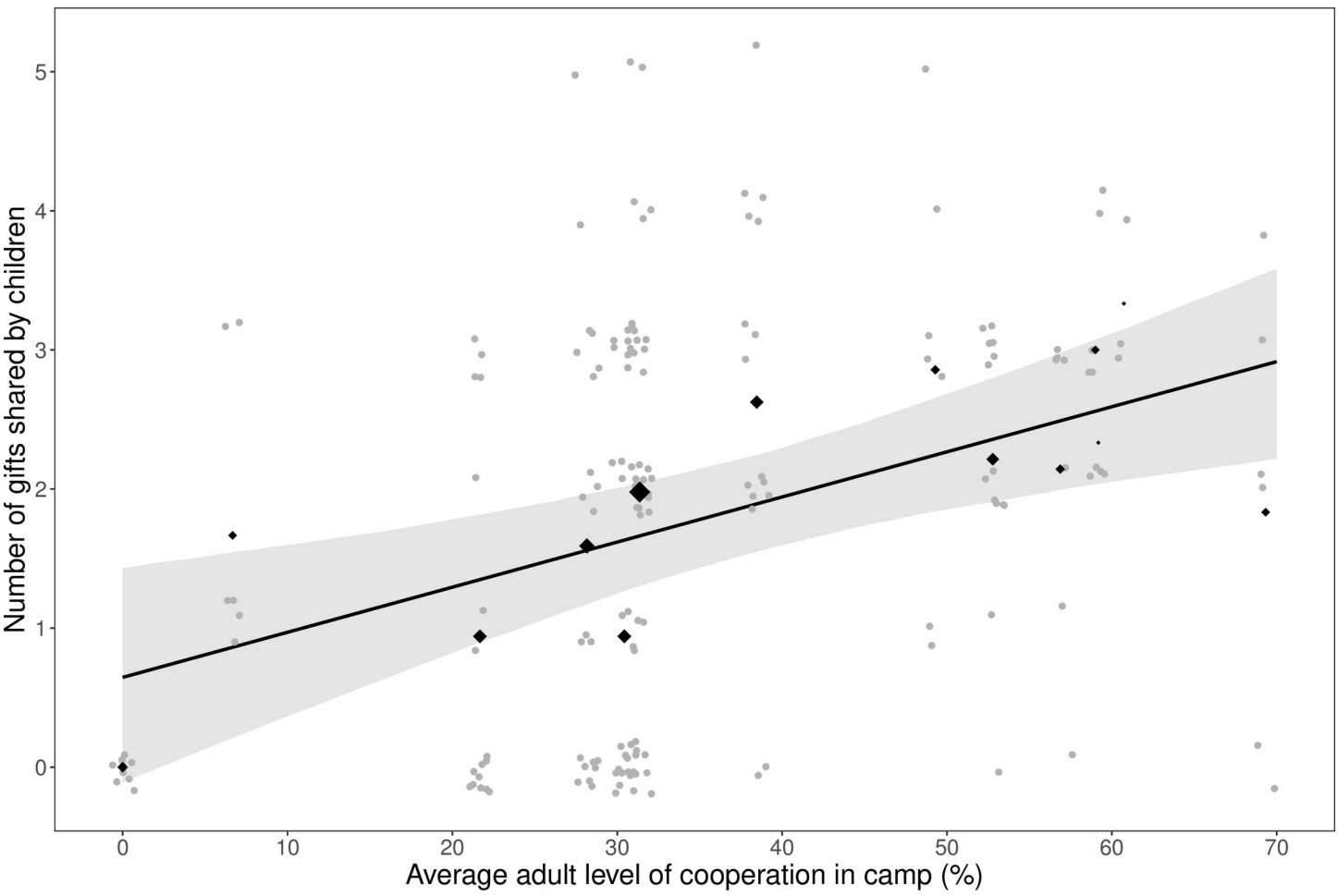

**Fig 2. Scatterplot displaying the relationship between child cooperation and average adult levels of cooperation in camp as measured in a giving game ($n$ = 179; camps = 14).** Grey circles indicate individual data points (with jitter to avoid overlap). Black diamonds indicate the average levels of cooperation per camp, with the size of the diamond representing the children's sample size in said camp. The regression line displays the predicted values of the amount shared from the univariable regression model with average camp-level adult cooperation as the predictor variable (95% credible interval in grey).

### Sharing partners

The next set of analyses explored the patterns of who children shared resources with, beginning with whether children preferentially shared with kin. The average relatedness between participant and recipient was $r$ = 0.30 (SD = 0.22), while the average relatedness to all child camp-mates was $r$ = 0.09 (SD = 0.05; S8 Fig in S1 File). A linear mixed-effects model with relatedness as the outcome found that relatedness between participant and recipient was significantly higher than expected by chance, with recipients possessing a relatedness coefficient with participants 0.21 greater than participant's average relatedness to other children in camp (95% CI = [0.18; 0.24]). Given the highly-skewed distribution of the data and residuals, both the assumptions of normality and homoskedasticity were violated (S9 and S10 Figs in S1 File). However, as the magnitude of the effect is so large, this is unlikely to impact the conclusion that children preferentially shared with close kin.

Next, associations between relatedness and participant age and sex were explored. Age was negatively associated with relatedness ($b$ = -0.015, 95% CI = [-0.025; -0.004]), meaning that older participants were more likely to share with less-related children; a 10-year increase in age

**Table 3. Comparison of models assessing linear and non-linear age associations with children's cooperation, and whether this association varies by camp, using a linear multi-level model ($n = 179$, camps = 14).**

| Model description | Model specification | LOOIC (SE) |
|---|---|---|
| *Total amount shared* | | |
| Linear age fixed effect with camp-level random intercepts | numShared ~ age + (1 \| camp) | 610.3 (16.5) |
| Quadratic age fixed effect with camp-level random intercepts | numShared ~ age + age^2 + (1 \| camp) | 612.8 (16.4) |
| Linear age fixed effect with camp-level random slopes and intercepts | numShared ~ age + (age \| camp) | 609.2 (16.8) |
| Quadratic age fixed effect with camp-level random slopes and intercepts | numShared ~ age + age^2 (age + age^2 \| camp) | 612.1 (16.8) |
| *Number of unique recipients* | | |
| Linear age fixed effect with camp-level random intercepts | numRecipients ~ age + (1 \| camp) | 573.5 (17.2) |
| Quadratic age fixed effect with camp-level random intercepts | numRecipients ~ age + age^2 + (1 \| camp) | 575.4 (17.1) |
| Linear age fixed effect with camp-level random slopes and intercepts | numRecipients ~ age + (age \| camp) | 573.9 (17.8) |
| Quadratic age fixed effect with camp-level random slopes and intercepts | numRecipients ~ age + age^2 (age + age^2 \| camp) | 576.4 (18.0) |

Results are repeated for both total amount shared (upper) and number of unique recipients (lower). LOOIC = Leave-one-out information criterion. SE = Standard Error.

predicted approximately a 0.15 reduction in the coefficient of relatedness between participant and recipient (Fig 3). Sex was weakly associated with relatedness, with boys somewhat more likely to share with closer kin ($b = 0.064$, 95% CI = [-0.003; 0.129]). No interaction between age and sex was found ($b = -0.005$, 95% CI = [-0.027; 0.016]). Tests of normality and homoskedasticity based on the model with age and sex as main effects indicate that these assumptions were likely to be violated (S11 and S12 Figs in S1 File); additional analyses are presented in S5 Table in S1 File which demonstrate that these findings are robust to alternative model specifications.

Additional models are provided in the supplementary information (S6-S9 Tables in S1 File) exploring how participant age, participant sex and relatedness, and their interactions, were associated with: i) recipient age; ii) the age difference between participant and recipient; iii) recipient sex; and iv) whether participant and recipient were of the same sex or not. Briefly, these analyses found that: older children were more likely to share with other older children, donations to younger children were towards closer kin, and there was a same-sex bias in children's nominations.

## Discussion

These results demonstrate that: i) the only factor strongly associated with levels of children's cooperation among the Agta was the average level of adult cooperation in camp, with higher levels of adult cooperation predicting increased child cooperation; ii) when allowing for partner choice, there is little association between child age and levels of cooperation; iii) older children were more likely to share with a greater number of camp-mates, indicating that older children possess a wider personal sharing network; and iv) children preferentially shared with close kin (especially siblings), although this kin-bias decreased with age, as older children shared more with non-kin and less-related children. Together, these results provide an insight into the evolutionary and developmental roots of cooperation among Agta hunter-gatherer

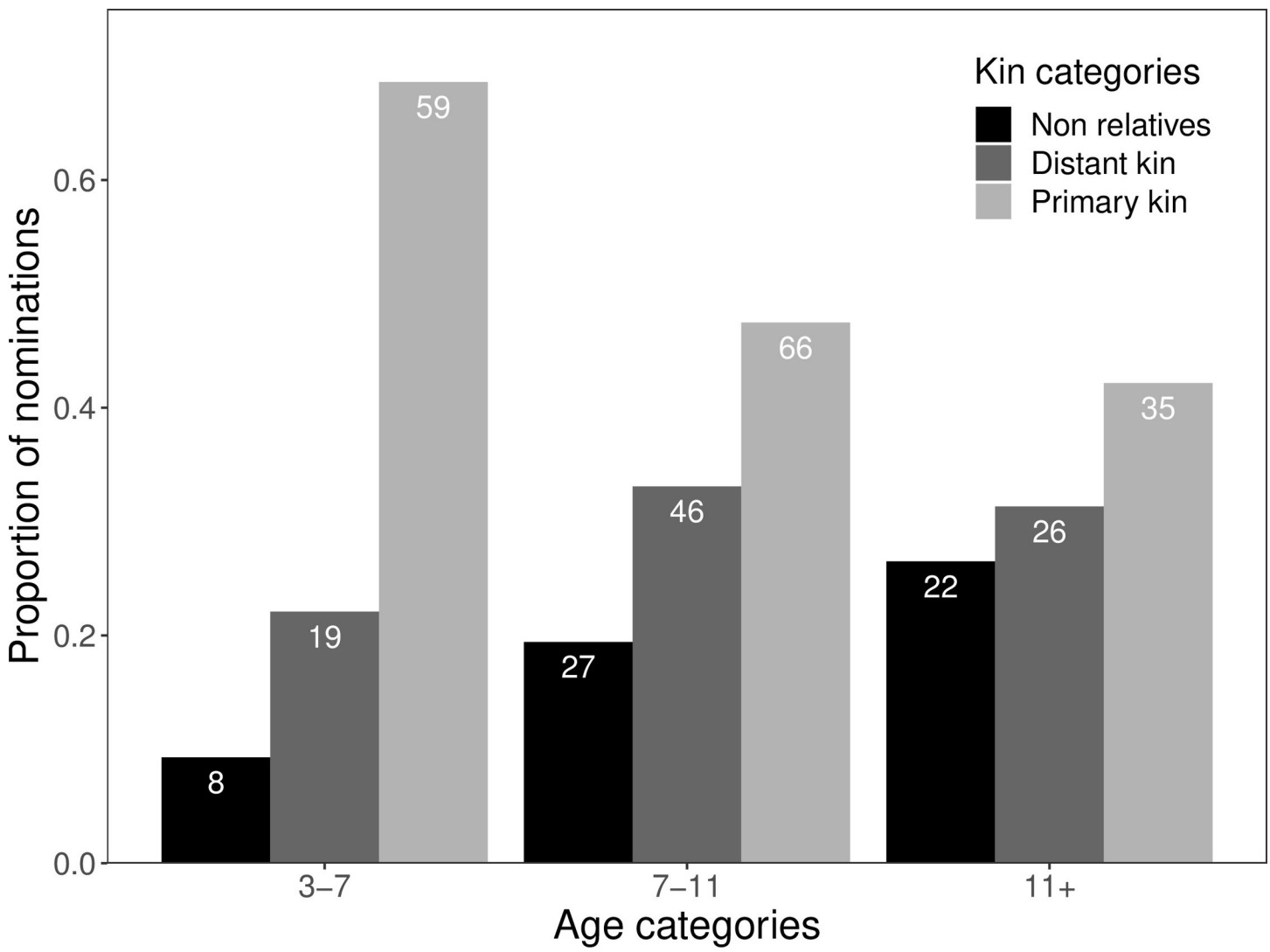

**Fig 3. Proportion of gifts given to either primary kin (light grey; *r* = 0.5), distant kin (dark grey; *r*< 0.5 but ≥0.03125; that is, all kin more closely related than second cousins, excluding primary kin) or non-relatives (black; *r*< 0.03125) as a function of age group.** Numbers in the bars indicate the total number of donations given to each kin category within each age group (*n* = 308 gifts from 125 children).

children, and may have implications for developmental studies of cooperation and the evolution of human childhood and life history more widely.

First, these results highlight the importance of assessing both levels of cooperation and partner choice simultaneously. Many previous studies which did not consider partner choice, and where cooperation was constrained to be between strangers or non-kin, found that cooperation increased with age (e.g., [11–13,15]). However, the present results suggest that this apparent increase in cooperativeness may be an artefact of these previous methodologies. When allowing for partner choice, Agta children between the ages of 3 and 18 were approximately equally cooperative, with perhaps only a marginal increase in cooperativeness with age. Further studies are of course required to explore whether the current findings replicate and how they are modified by the social, ecological and cultural environment, but we tentatively suggest that cooperation with *non-kin* (or more distant relatives) may increase with age, but not necessarily that cooperation *overall* increases over childhood. Thus, the low levels of cooperation of

younger children found in previous studies may be because they could not selectively choose kin as recipients, rather than being less cooperative *per se*.

While little effect of age was found regarding overall levels of cooperation, high levels of between-camp variation were reported. Consistent with the results from previous research across six societies [12], Agta children were more cooperative in camps where adults were more cooperative. However, the slope of this age association was similar across all camps, arguing against the idea that children acquire these camp-specific cooperative behaviours in middle childhood, as suggested by previous work [12,15]. The mechanism(s) underlying these patterns is unclear. As adult levels of cooperation were predicted by adaptive factors such as repeated interactions, resource availability and the presence of skilled storytellers [26,53], it is possible that similar adaptive considerations directly shape levels of children's cooperation as well. Alternatively, it may be that adult cooperation is responsive to these adaptive factors, but that children's cooperation is shaped by socially learning from adults in camp, rather than being sensitive to the varying costs and benefits of cooperation under different socioecological conditions. Answering this important question is beyond the scope of the present study. None-theless, regardless of the mechanism, it does appear that Agta children are behaving in ways broadly consistent with adaptive considerations, even from an early age. One possibility which can appear to be ruled out, however, is simple vertical transmission of cooperative tendencies from parents to children. There was no evidence that parental levels of cooperation were associated with children's levels of cooperation when controlling for between-camp differences in cooperation. Additionally, while previous developmental research (e.g., [12,13,15]) only explored variation at the ethnolinguistic level, here we demonstrate that similar variation can be observed within groups, meaning that we cannot assume homogenous cooperation in children at the cultural group level [26,55].

Regarding patterns of sharing partners, we find that most cooperation is directed towards siblings (52% of all gifts) although in later childhood sharing becomes less nepotistic as children increasingly share with distant-kin and non-relatives. We also find that the size of children's personal sharing networks increases with age, as older children share with a greater number of recipients. Thus, although overall levels of cooperation vary little with age, who children cooperate with, and the demographic composition of their personal sharing networks, changes considerably over childhood. Regardless of the specific mechanisms by which Agta children acquire these cooperative behaviours, patterns of cooperation among Agta children are comparable to those among Agta adults, with considerable variation in cooperativeness between camps and a strong kin bias in sharing partners [26,56,57].

What are the potential functions of these patterns and what are their implications for human life history? First, these results demonstrate that Agta children willingly share resources with siblings, especially younger siblings. This supports existing literature highlighting the importance of siblings as alloparents, broadly defined here as non-parental childcare (for a nuanced discussion, see [58]). This matches real-world patterns of cooperative childcare, where much childcare is undertaken by older siblings [7,9,37], and that this investment may have important fitness consequences [59,60]. This suggests that even young children may invest in younger siblings for indirect fitness benefits, perhaps potentially at a cost to their direct fitness. Consistent with this idea, recent studies in UK and Czech cohorts found that delayed menarche–a proxy for age at reproduction–was associated with having full siblings, relative to having only half- or step-siblings [61,62]. This suggests that individuals may delay reproduction–a direct fitness cost–to assist close kin. Siblings can of course compete for resources [63], but overall the current results are consistent with the idea that aspects of our derived life history, such as our extended childhood period, may in part be due to our system of cooperative childrearing where help is preferentially directed towards full siblings [64].

However, as some cooperation, especially in older children, was increasingly directed towards distant kin and non-relatives, indirect fitness benefits cannot be the sole explanation for why children cooperate. As children get older, they become increasingly involved in wider camp and society life, necessitating more extensive interactions and cooperation with less-related individuals [30–33]. Speaking very broadly, these results suggest that cooperation in early childhood may function to assist in the care of siblings for indirect fitness benefits, while in later childhood individuals increasingly cooperate with non-kin peers to forge friendships and learn the necessary skills for navigating adult social life for direct fitness benefits. It is important to note that even in older children ~40% of all gifts were given to siblings, suggesting that sibling care is still an important consideration at these ages. It would be extremely enlightening to see whether similar patterns are found in other societies using a comparable methodology.

A key limitation of this research is that cooperation is measured via experimental games, meaning that is it not clear how this links to real-world cooperative behaviours [65]. Nonetheless, as behaviour in similar games appeared to reflect real-world food-sharing practices among adult Agta [26], this provides some support that game behaviour may reflect real-world cooperative behaviour, at least regarding food-sharing. Additionally, many criticisms about the validity of experimental games apply to methods with anonymous recipients; by employing methods with non-anonymous recipients here, it is hoped that many of these concerns will be addressed and results more reliable [40,41]. Further bolstering the validity of these findings, the observed patterns of partner choice reflect changes in proximity network patterns over childhood, from close kin when young to more distant and non-kin in later childhood [28], demonstrating that behaviour in these games reflects children's real-world interaction networks. The current results also highlight the importance of kinship in shaping patterns of children's childcare, as has been found in many other populations [7,23], including the Agta [24]. This indicates that the patterns of results from this simple resource allocation game are congruent with existing empirical work on human alloparenting. These findings are therefore consistent with multiple independent lines of evidence, providing assurance that these results possess some degree of external validity.

A further potential limitation is that there were no formal comprehension tests to ensure that children understood the game, other than simply asking if they understood. Given that the design of the study was intended to be as simple as possible, and there appeared to be no issues in comprehension during the piloting of this design, it was felt that formal comprehension tests were not necessary. It is our judgment that the vast majority of children did understand the game and acted accordingly. Nonetheless, this subjective judgement may be incorrect, and it is possible that a lack of comprehension may have biased these results. If children–and young children especially–did not understand the task, then one may expect their behaviour to be largely random; instead, we observed large camp-level differences in cooperation, even among younger children. This suggests that we observed these patterns of results perhaps *in spite of* potential comprehension issues, which would be expected to result in reduced power to detect a true effect. Although no obvious issues in comprehension were apparent, we acknowledge this as a potential limitation and recommend that future research using similar methodologies should employ formal comprehension checks.

An additional potential limitation is the design of study for testing theories of partner choice. As children were simply asked to think of potential recipients, children of different ages may differ in who they could think of to share with. For instance, as young children interact less with non-kin [28], they may not have considered these individuals as potential recipients, meaning that the kin-bias observed at early ages may have been a methodological artifact. While this interpretation is possible, we believe it is unlikely for a number of reasons: i) this

decreasing kin-bias with age is consistent with theoretical expectations (as discussed above); ii) patterns of partner choice in this game correspond to observed interaction networks among Agta children [28], providing a measure of external validity; and iii) to the best of our knowledge, there are no studies comparing children's partner choice decisions both with and without identity prompts, meaning the impact of our methodology on children's decisions–if any–is unknown. We do however acknowledge this as a potential limitation, and additional research is needed, especially comparing methods with vs without identity prompts and exploring whether the patterns of sharing partners observed here replicate in other populations.

Given that low numbers of children participated in certain camps–two camps only had three participants, while eight of the 14 camps included less than 10 children–sample size and statistical power are important potential limitations. When working with small-scale populations these issues are unfortunately frequently unavoidable, but this does mean that the current analyses may be underpowered to detect weak or camp-specific effects. Additionally, although some models violated the assumptions of normality and homoskedasticity, these violations are unlikely to lead to biased effect estimates [66]. Nonetheless, where possible, alternative model specifications, which do not rely on these assumptions, found qualitatively similar patterns of results, suggesting that these conclusions are robust.

The observed lack of increased cooperation with age, coupled with marked between-camp differences in cooperativeness and highly-structured patterns of partner choice, suggest that childhood may not simply be a time to acquire adult norms. Rather, children's behaviour can be understood as (potentially) adaptive in its own right, either because it benefits them in the present, or is necessary for developing future social or cultural skills [33,34]. That is, when interpreting developmental studies it is important to distinguish between *ultimate/functional* explanations, in terms of why a behaviour evolved and whether it is, or recently was, adaptive, and *proximate* explanations based on development (how a behaviour is acquired over ontogeny) or mechanism (how a behaviour works; [4,67]). Much developmental work on cooperation focuses on the latter *how* questions–*how, from whom*, and *when* do children learn cooperative behaviour?–especially in terms of cultural transmission (e.g., [12,13,68]), without necessarily asking *why* these patterns exist and whether said behaviour is adaptive [1,69,70]. While children likely acquire much of their cooperative behaviour through social learning, this social learning likely reflects fitness-enhancing considerations [71,72]. Acquiring these adaptive behaviours may also be explicable in terms of non-social mechanisms, such as individual learning or reaction norms. From the data here it is impossible to assess the relative contributions of these proximate mechanisms, although a simple parent-offspring vertical transmission pathway can be ruled out. Nonetheless, the observed patterns are broadly consistent with a functional interpretation, regardless of the mechanistic pathway. Variation in levels of children's cooperation between camps are structured in broadly-adaptive ways, such as greater cooperation in more stable camps. Patterns of partner choice are also consistent with an adaptive function of helping siblings in early childhood for indirect fitness benefits, followed by increased cooperation and integration with the wider camp in later childhood for direct fitness benefits. While these specific conclusions are tentative and require replication, this study has hopefully demonstrated the potential benefits of taking an adaptive evolutionary approach to understanding the development of children's cooperative behaviour.

## Supporting information

**S1 File. Supplementary information for "cooperation and partner choice among agta hunter-gatherer children: An evolutionary developmental perspective".** This

supplementary information file contains additional information regarding the aging method (section S1), additional details of the children's cooperative game methods (section S2), a summary of the methods for the cooperative games played with Agta adults (section S3), plus all supplementary tables (S1-S9 Tables) and figures (S1-S16 Figs).
(PDF)

## Acknowledgments

We thank the Agta communities and research assistants for their help and hospitality in the field and for making this research possible. Thanks also to Fiona Jordan and EXCD.lab team at the University of Bristol, the London School of Hygiene and Tropical Medicine Evolutionary Demography group, and Bailey House for helpful feedback on earlier versions of this work.

## Author Contributions

**Conceptualization:** Daniel Major-Smith, Ruth Mace, Andrea B. Migliano.

**Data curation:** Daniel Major-Smith.

**Formal analysis:** Daniel Major-Smith.

**Funding acquisition:** Ruth Mace, Andrea B. Migliano.

**Investigation:** Daniel Major-Smith, Mark Dyble, Katie Major-Smith, Abigail E. Page.

**Methodology:** Daniel Major-Smith, Andrea B. Migliano.

**Project administration:** Ruth Mace, Andrea B. Migliano.

**Supervision:** Ruth Mace, Andrea B. Migliano.

**Visualization:** Daniel Major-Smith.

**Writing – original draft:** Daniel Major-Smith.

**Writing – review & editing:** Daniel Major-Smith, Nikhil Chaudhary, Mark Dyble, Katie Major-Smith, Abigail E. Page, Gul Deniz Salali, Ruth Mace, Andrea B. Migliano.

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
