## [Decision Letter · Decision Letter 0]

26 Dec 2022

PONE-D-22-29549Cooperation and Partner Choice Among Agta Hunter-Gatherer Children: An Evolutionary Developmental PerspectivePLOS ONE

Dear Dr. Major-Smith,

Thank you for submitting your manuscript to PLOS ONE. After careful consideration, we feel that it has merit but does not fully meet PLOS ONE’s publication criteria as it currently stands. Therefore, we invite you to submit a revised version of the manuscript that addresses the points raised during the review process. Please submit your revised manuscript by Feb 09 2023 11:59PM. If you will need more time than this to complete your revisions, please reply to this message or contact the journal office at plosone@plos.org. Please include the following items when submitting your revised manuscript:A rebuttal letter that responds to each point raised by the academic editor and reviewer(s). You should upload this letter as a separate file labeled 'Response to Reviewers'.A marked-up copy of your manuscript that highlights changes made to the original version. You should upload this as a separate file labeled 'Revised Manuscript with Track Changes'.An unmarked version of your revised paper without tracked changes. You should upload this as a separate file labeled 'Manuscript'.

We look forward to receiving your revised manuscript.

Kind regards,

Daniel Redhead

Academic Editor

PLOS ONE

Journal Requirements:

   "This work was supported by the Leverhulme Trust (grant no. RP2011 R045; awarded to ABM and RM). DM-S was also supported by the John Templeton Foundation (grant ID: 61917)."

Additional Editor Comments:

Two reviewers with expertise in the field and I have now read the manuscript. We all agree that the manuscript presents interesting findings and addresses important questions on the development of cooperative behaviour over childhood. While we all see great promise, both reviewers and I believe that major revisions to methods and presentation would greatly improve the manuscript. Both reviewers have been clear and detailed in their feedback, so I will not labour their points by repeating them here. I will, however, outline the major points for the revision that the reviewers and I converge on.

- Theoretical motivation: As reviewer 1 notes, the introduction would be improved by incorporating more relevant existing research on social learning. Alongside this, I think that the theoretical justifications for predictions would be strengthened by drawing from relevant previous research on real-world adult cooperation in `small-scale’ societies.

- Statistical analyses: Both reviewers raise concerns about the analyses. The study was in-part motivated by network theory, and the data collected here are network structured. Therefore, it was surprising to see that no network methods were implemented to analyse such data. Given the dependence structure of such data—and the research questions—it seems like the most appropriate analytical strategy would be to develop/apply statistical models of network structure (e.g., social relations models as reviewer 1 notes, or for example appropriately specified ERGMs) would be the most appropriate. For instance, relatedness could be included as a dyadic covariate, and sender and receiver effects could be estimated for gender in a social relations model. Moreover, predictions on the association between age and network size could be computed an visualised from that same model. As reviewer 1 notes, the current description of the analytical strategy and results are also not super clear. In a revision to the manuscript, I would really like to either a) see network methods being implemented for the analyses, or b) a strong justification for, and restructuring and clear explanation of, the current methods.

- General presentation: The manuscript would be strengthened by some further thorough editing. I agree with review 2, who notes that more concision and further grammatical corrections (e.g., shorter sentences with clear subjects) would make the manuscript more readable.

Alongside these major concerns, please respond to all of the points made by reviewers in your revision.

Thank you again for submitting this really interesting study for publication at Plos One, and I hope that you find our feedback helpful.

Reviewers' comments:

Reviewer's Responses to Questions

**Comments to the Author**

1. Is the manuscript technically sound, and do the data support the conclusions?

Reviewer #1: Partly

Reviewer #2: Partly

2. Has the statistical analysis been performed appropriately and rigorously? 

Reviewer #1: No

Reviewer #2: No

3. Have the authors made all data underlying the findings in their manuscript fully available?

Reviewer #1: Yes

Reviewer #2: Yes

4. Is the manuscript presented in an intelligible fashion and written in standard English?

Reviewer #1: Yes

Reviewer #2: No

5. Review Comments to the Author

Reviewer #1: See PDF. See PDF. See PDF. See PDF. See PDF. See PDF. See PDF. See PDF. See PDF. See PDF. See PDF. See PDF. See PDF. See PDF. See PDF. See PDF. See PDF. See PDF. See PDF. See PDF. *Remove character count check from this field PLoS.

Reviewer #2: This is an interesting paper that investigates the level and pattern of cooperative behaviours of Agta children over the developmental period, using a simple resource allocation game. Although many other experimental studies have explored how cooperative behaviours emerge in children and increase over childhood, this study provides additional details and identifies with whom children are likely to share. Surprisingly, only the average level of cooperation among adults in camp had the effect on the level of cooperation of children. Age, sex, or within-camp average relatedness did not have any effects how much children shared with others. Regarding sharing partners, children were more likely to share with close kin, and children increasingly shared with less-related individuals as they aged.

This work presents an experimental dataset investigating how much and with whom children share the given resources in a hunter-gatherer population. The result supports one of well-known propensities of human cooperation: the tendency to increase the benefit of close relations, and further provides an evidence that children’s cooperative networks increases with age, from kin to distant kin and non-kin. I believe that it is potentially a very important and insightful paper.

However, there are a few points which should be addressed before further assessment. I have a few main concerns with this paper. First, the predictions are not clearly justified theoretically especially in the Introduction, some of them were discussed later only in Discussion. Second, the authors need to be more careful when interpreting the results and formulating sentences throughout the paper—specifically about partner choice. Third, the paper needs to be re-organised to be clear in explaining the statistical analysis methods and results, as with the current state it is hard to follow what the authors did for statistical analyses and, thus, it is hard to draw the main findings and interpretation the authors claimed that they found. Lastly and most importantly, the authors should use alternative methods to solve several convergence issues that they faced during analyses. I also believe that the paper needs substantial revision on their writing. to See more details ‘Major issues’ below.

Major Issues

1. This paper aims to investigate cooperative behaviours of children from developmental perspectives, but the predictions do not clearly provide theoretical reasoning for these predictions from developmental perspectives. The main predictions are that the level of cooperation would increase with age (L 129) and that children’s level of cooperation would vary between camps and will correspond to the adults’ patterns of cooperation in each camp (L 130-133). In the current text, it reads like the authors set the predictions because previous research found the same pattern, and therefore they also expect to find the same pattern. Explaining clear reasoning for predictions which stands on theory in the Introduction will make this paper interesting. Developmental research has demonstrated that children are sensitive to social norms, and that and that social norms are internalised by middle childhood. Social learning research has shown that that social learning mechanisms contribute to children’s acquisition of community-specific social norms, including sharing norms. This study can add to this body of research by investigating developmental trajectories of sharing behaviours. However, social learning is discussed only later in the Discussion and social norms are not mentioned. I think it is important to bring theory in Introduction when building the predictions.

2. The second research question and prediction the authors are investigating is how partner choice shapes children’s cooperative behaviour (L 90, L 98). However, this work did not explore how partner choice shapes cooperation, but is only examining who are sharing partners and how this change over childhood.

3. The Method and Result sections need quite a bit of re-organisation to make the paper clearer. The results section is hard to follow because the explanations on models are mixed and the organisation is repetitious, thus, hard to understand what the main findings are and to draw the interpretation. Instead of repeating the model structures in the Results section, I suggest the authors to explain statistical model structures more clearly in the Methods part, probably by using a table with model structures or by bringing a DAG figure to the main text (which is only in Supplementary Information), and to start the Results with clear descriptions of their findings. For example, model explanations (L 326 – L 333, L 403 – 409, L 427 – 430, L 436 – L440, L441 – 444) in the Results can be moved into the statistical analysis part and deleted from here.

4. My biggest concern regards the statistical analysis methods. The authors faced several convergence issues during statistical analyses and they noted that therefore they used similar models (L 249, L 299, L 367, L 604). Yet there are many analysing methods which can solve convergence issues, for example, by using Bayesian modelling (also noted by the authors in Line 607). I wonder why the authors would not use Bayesian modelling, for example, using ‘brms’ package in R. Moreover, in Line 424 and Line 438, the authors wrote that the assumptions of normality and homoskedasticity were violated due to highly skewed distribution of the data. Here I am not convinced how this is not a problem (L 425) and whether those model results can be interpreted as it is in the paper now.

5. The paper does not read clearly and smoothly. The authors need to increase clarity and brevity of the paper, while avoiding much repetition. Especially, the Discussion is very hard to follow and thus to buy their claims. For example, in Discussion the authors acknowledged the limitation of the study over several paragraphs. It is good that the authors brought up the limitation of the study, addressed their points, and suggested the future directions. However, those points were not addressed well in scientific writing style— for example, Line 548, “This study of course possesses limitations. Chief among these is that cooperation is measured via experimental games, meaning that is it not clear how this links to real-world cooperative behaviours.” This sentence should be formulated again. Moreover, despite the valuable contribution of this study, the authors’ voice became too week after explaining the limitation to buy their claims. This is clearly shown in the last sentence of the last paragraph of the paper (from Line 652), saying “While these specific conclusions are tentative and rather speculative, this study has hopefully demonstrated the potential benefits of taking an adaptive evolutionary approach to understanding the development of children’s cooperative behaviour.”, which is quite different from their voice in the last sentence of the first paragraph of the Discussion (Line 464-467), saying “Together, these results provide an insight into the evolutionary and developmental roots of cooperation among Agta hunter-gatherer children, and have implications for developmental studies of cooperation and the evolution of human childhood and life history more widely.” With this varying tone and voice, it is hard for readers to believe the conclusion of this study. The paper needs substantial revision on their writing style.

Minor Issues

1. I would include ‘sharing’ explicitly throughout the text, so that the paper can read more clearly: for example, in title “Cooperation and sharing partners among…”

2. Regarding partner choice, the authors can define it more clearly what it means in this study -- sharing partners.

3. Throughout the manuscript, the authors can use brackets “( )” a lot to explain something. I suggest to avoid using brackets too much, and rather write it out as a full sentence.

4. In Ethnographic background, I suggest the authors to describe the structure of each camp, what varies (relatedness, number, sex ratio) and how it varies, as camp variation is one of the important factors influencing the results.

5. In Data collection, the authors could explain the game procedure more clearly, especially Line 201-204. Does “after all games were played” mean “after all children participated in the game”? During the game, the participant did not know that they will anyway get candies later, regardless of the game results? This thing can be explained more clearly.

6. In Statistical analysis, it is not so clear which models the authors ran, which models failed due to convergence issues, and which results from which model the authors use for interpretation.

7. The subtitles in the Result section: “Amount shared” and “Partner Choice Analyses” can be changed “The level of sharing” and “Sharing partners”.

8. In Result, Table 1 is not intuitive and not easy to understand. I suggest to separate columns, one for Number shared, and another for Number of unique recipients.

9. In Discussion, I suggest the authors to discuss Agta sharing norms among adults, for example regarding generalised reciprocity and kinship, as well as to add some description on food sharing norms in Agta’s daily lives.

Line 37: Change “cooperative development” to “development of cooperative behaviours” for clarity.

Line 84: Change “in that” into “in which”.

Line 181: Change “to assess cooperation” into “to assess the level and pattern of cooperation”.

Line 339: Where did these two additional resources come from?

Line 464-467: I would move to the last paragraph, and here I would give more specific summary of what this finding means.

6. PLOS authors have the option to publish the peer review history of their article (what does this mean?). If published, this will include your full peer review and any attached files.

Reviewer #1: No

Reviewer #2: No

---

## [Author Response · Author response to Decision Letter 0]

1 Feb 2023

Please see the attached 'response to reviewers' PDF.

---

## [Decision Letter · Decision Letter 1]

22 Mar 2023

PONE-D-22-29549R1Cooperation and partner choice among Agta hunter-gatherer children: an evolutionary developmental perspectivePLOS ONE

Dear Dr. Major-Smith,

Thank you for submitting your manuscript to PLOS ONE. After careful consideration, we feel that it has merit but does not fully meet PLOS ONE’s publication criteria as it currently stands. Therefore, we invite you to submit a revised version of the manuscript that addresses the points raised during the review process.

The revised manuscript now reads much clearer and the new analyses seem much more appropriate. Both the reviewer and I believe that only minor revisions are necessary for the manuscript to be of an acceptable standard for publication. Please address all of the minor comments made by the reviewer and myself.

We look forward to receiving your revised manuscript.

Kind regards,

Daniel Redhead

Academic Editor

PLOS ONE

Journal Requirements:

Additional Editor Comments:

Here are a handful of more specific issues that I think need addressing in the revision:

Methods

- How was the 'Sex' variable constructed? Was it using self-report data? If so, it might be important to consider using the term 'gender'. If not, it would be great to state why 'sex' is used. 

- 250-251: I'm guessing that Wright's coefficient of relatedness is used here? How was it calculated, using the kinship2 package? If so, please note this and cite the package. 

Discussion

Generally, I think that the discussion of results around children's network size and structure could be toned down a little, and use of terminology slightly more refined. First, I think it would be best to add "personal" in front of "networks". This is a minor change, but has important consequences for readers. The present analyses is essentially capturing a partial sample of a network or individuals' ego networks. Therefore, the present data/analyses cannot directly speak to the global structure of children's networks in the population. Given this, it might be considered quite a stretch to conclude that the present findings show that "the structure of [children's] cooperation networks, changes considerably over childhood" (lines 552-554). The structure of children's networks is not something that is being examined here, and this quite conclusive statement begs questions as to what structural properties of a network are observed to change. It seems that the results might speak more directly to the demographic composition of children's cooperation partners being different across ages. 

Reviewers' comments:

Reviewer's Responses to Questions

**Comments to the Author**

1. If the authors have adequately addressed your comments raised in a previous round of review and you feel that this manuscript is now acceptable for publication, you may indicate that here to bypass the “Comments to the Author” section, enter your conflict of interest statement in the “Confidential to Editor” section, and submit your "Accept" recommendation.

Reviewer #2: All comments have been addressed

2. Is the manuscript technically sound, and do the data support the conclusions?

Reviewer #2: Yes

3. Has the statistical analysis been performed appropriately and rigorously? 

Reviewer #2: N/A

4. Have the authors made all data underlying the findings in their manuscript fully available?

Reviewer #2: Yes

5. Is the manuscript presented in an intelligible fashion and written in standard English?

Reviewer #2: Yes

6. Review Comments to the Author

Reviewer #2: I am glad to read the revised manuscript, as the authors improved their statistical analysis methods more rigorously, and it reads well. It is a solid and interesting paper which will contribute to this field.

My one remaining concern is the model results of “Sharing Partner”. The model results of “Amount Shared” are clearly found in Table 2, but I cannot find the main result tables for “Sharing Partner”, also not in the supplementary information either, if I understood correctly. In Lines 466 , 473, 477, 478, authors showed the estimates and 95% confidence intervals, where can the readers find these results in the model tables? I found only Table S5-S9 for additional analyses, but not the main results. In general, compared to the “Amount Shared”, description of analysis methods and results of “Sharing Partner” are not clear. Hence, I recommend the authors to add a model result table of sharing partner in the main manuscript.

And below I have some more minor comments to make the paper clearer.

Line 69: Authors can add reference Jang et al., 2022 on children help mothers increase food returns during foraging.

Jang, H., Janmaat, K. R., Kandza, V., & Boyette, A. H. (2022). Girls in early childhood increase food returns of nursing women during subsistence activities of the BaYaka in the Republic of Congo. Proceedings of the Royal Society B, 289(1987), 20221407.

Line 75-76: Could you explain what patterns of cooperation are similar across societies until middle childhood?

Line 80: Authors can add reference Lew-Levy et al., 2018, a review paper on how hunter-gatherer children learn social norms.

Lew-Levy, S., Lavi, N., Reckin, R., Cristóbal-Azkarate, J., & Ellis-Davies, K. (2018). How do hunter-gatherer children learn social and gender norms? A meta-ethnographic review. Cross-Cultural Research, 52(2), 213-255.

Line 83: Add reference (11) in the end of sentence.

Line 92: Change “For instance, say that young children may preferentially only share….” to “For instance, young children may preferentially only share….”.

Line 115: Change “Partner choice is unlikely to…” to “The pattern of partner choice is unlikely to…”.

Line 123: Authors can add references Lew-Levy et al., 2017; 2018, two review papers on how hunter-gatherer children learn social norms and subsistence skills.

Lew-Levy, S., Reckin, R., Lavi, N., Cristóbal-Azkarate, J., & Ellis-Davies, K. (2017). How do hunter-gatherer children learn subsistence skills? A meta-ethnographic review. Human Nature, 28, 367-394.

Lew-Levy, S., Lavi, N., Reckin, R., Cristóbal-Azkarate, J., & Ellis-Davies, K. (2018). How do hunter-gatherer children learn social and gender norms? A meta-ethnographic review. Cross-Cultural Research, 52(2), 213-255.

Line 136: This sentence is repetitive with previous paragraphs, the authors can maybe shorten this paragraph and merge it with the previous paragraph, probably by saying: “Therefore, to compare overall levels of cooperation as well as to explore who children share with and whether this pattern changes with age, incorporating partner choice into experimental designs is necessary. While partner choice has been increasingly investigated among adults (23, 36-39), there are only few studies applying such methods with children, especially in small-scale societies.”.

Line 153: Authors can change “….adult-levels of cooperation (previous research…… (24))” to, “….adult-levels of cooperation, as previous research…… (24)”.

Line 178: Can we still call the camp with 26 houses with 156 individuals as camp, but not a village?

Line 217: Authors can change “…both of these are decisions made repeatedly by children every day….” to “…both of these are decisions made repeatedly by children in their daily lives….”

Line 224: Authors can probably delete “after which the game was played”? Did authors ask these questions also before the game was played?

Line 233: Authors can change to “too shy and/or too young to fully understand”.

Line 257: Authors can change to “the causal relations between variables were considered”.

Line 270: Authors can change to “Our hypothesised causal structure suggest that…..”.

Line 275, 375: Using the terminology “null linear model” or “null multi-level model” can make readers confused with null hypothesis testing, so I would suggest to use “a baseline model which accounts for multilevel structure of data with random effects, but includes no covariates” instead of “null model” throughout the paper.

Line 276: “camp ID” and “household ID” for precision.

Line 301: “The first set of analyses aimed to assess whether….”.

Line 313: Do authors mean “When only one of the parents, either mother or father, participated, the score from the one-parent was used as an average score in the combined parental model.”?

Line 314: Authors investigated specifically how children shared candies to others, therefore, I would suggest using “sharing networks” instead of “cooperative networks” throughout the paper.

Line 327: How can authors still have “major convergence issues”?

Line 365: To be more accurate, change to “participant’s age, sex and participant’s relatedness to recipient”.

Line 390-396: This paragraph can still be moved to the end of Statistical Analyses part for Amount Shared.

Table 3: This can be moved to the Supplementary Information.

In Model specification in Table 3, “childCoop” can be changed to “numShared” to increase accuracy.

Line 525: “Consistent with the result from previous research across six societies (11),”

Line 529: “middle childhood”, instead of “mid-childhood”

Line 553: I would avoid using the term “the structure of their cooperative networks”, especially as the authors clearly wrote that they did not use social network analyses, and I am also not convinced which network structure they found.

Line 608-623: I think this paragraph can be potentially removed as the authors made sure the children understood the games (see Line 224).

7. PLOS authors have the option to publish the peer review history of their article (what does this mean?). If published, this will include your full peer review and any attached files.

Reviewer #2: No

---

## [Author Response · Author response to Decision Letter 1]

24 Mar 2023

Please see the attached 'response to reviewers' document.

---

## [Editor Report · Decision Letter 2]

29 Mar 2023

Cooperation and partner choice among Agta hunter-gatherer children: an evolutionary developmental perspective

PONE-D-22-29549R2

Dear Dr. Major-Smith,

We’re pleased to inform you that your manuscript has been judged scientifically suitable for publication and will be formally accepted for publication once it meets all outstanding technical requirements.

Kind regards,

Daniel Redhead

Academic Editor

PLOS ONE

Additional Editor Comments (optional):

The authors have addressed all issues raised by the editor and reviewers, and the manuscript is now in an acceptable condition for publication. Great work!
---

## [Editor Report · Acceptance letter]

3 Apr 2023

PONE-D-22-29549R2 

Cooperation and partner choice among Agta hunter-gatherer children: an evolutionary developmental perspective 

Dear Dr. Major-Smith:

I'm pleased to inform you that your manuscript has been deemed suitable for publication in PLOS ONE. Congratulations! Your manuscript is now with our production department. 

Kind regards, 

on behalf of

Dr. Daniel Redhead 

Academic Editor

PLOS ONE